# The Current State of Tumor Microenvironment-Specific Therapies for Non-Small Cell Lung Cancer

**DOI:** 10.3390/cancers17111732

**Published:** 2025-05-22

**Authors:** Raghav Chandra, Jasmina Ehab, Edward Hauptmann, Naga Swati Gunturu, John D. Karalis, Daniel O. Kent, Christopher A. Heid, Scott I. Reznik, Inderpal S. Sarkaria, Huocong Huang, Rolf A. Brekken, John D. Minna

**Affiliations:** 1Department of Surgery, University of Texas Southwestern Medical Center, Dallas, TX 75390, USA; raghav.chandra@utsouthwestern.edu (R.C.); edward.hauptmann@utsouthwestern.edu (E.H.); naga.gunturu@utsouthwestern.edu (N.S.G.); john.karalis@utsouthwestern.edu (J.D.K.); huocong.huang@utsouthwestern.edu (H.H.); rolf.brekken@utsouthwestern.edu (R.A.B.); 2Department of Surgery, Beth Israel Deaconess Medical Center, Harvard Medical School, Boston, MA 02115, USA; jasmina.ehab@utsouthwestern.edu (J.E.); dkent@bidmc.harvard.edu (D.O.K.); 3Department of Cardiovascular and Thoracic Surgery, University of Texas Southwestern Medical Center, Dallas, TX 75390, USA; christopher.heid@utsouthwestern.edu (C.A.H.); scott.reznik@utsouthwestern.edu (S.I.R.); inderpal.sarkaria@utsouthwestern.edu (I.S.S.); 4Hamon Center for Therapeutic Oncology Research, University of Texas Southwestern Medical Center, Dallas, TX 75390, USA; 5Department of Internal Medicine and Pharmacology, University of Texas Southwestern Medical Center, Dallas, TX 75390, USA

**Keywords:** NSCLC, tumor microenvironment, targeted therapy, immunotherapy, thoracic surgical oncology

## Abstract

The tumor microenvironment (TME) contributes to all aspects of tumor progression in non-small cell lung cancer (NSCLC). The advent of immunotherapy has revolutionized the care for patients with advanced disease. In this review, we highlight the current preclinical and clinical investigations into therapies targeting the structural and immune-cellular components of the NSCLC TME. We further explore the emerging role of perioperative immunotherapy and TME-specific considerations for early-stage and potentially resectable disease.

## 1. Introduction

Non-small cell lung cancer (NSCLC) continues to be the leading cause of cancer-related mortality worldwide, with an estimated 1.8 million deaths in 2022 [1]. Unresectable disease carries a grave prognosis, with a median 5-year survival of 18% [2]. Our understanding of NSCLC has now expanded beyond the primary cancer cells into the functions of, and therapeutic targets in, the tumor microenvironment (TME). As the eminent British surgeon Stephen Paget first described in his “seed and soil” theory in 1889, the TME’s individual components, including structural cells, endothelial cells, and immune cells, contribute meaningfully to virtually all hallmarks of cancer progression, including metastasis, angiogenesis, therapy resistance, and stemness [3,4]. Fortunately, remarkable advancements have been made in the past decade with respect to our understanding of the TME and targeting it for novel therapeutic strategies. Chief among them has been the landmark development of “immune checkpoint inhibitor” (ICI) immunotherapy using antibodies targeting PD-L1, PD-1, and CTLA-4, whose indications have expanded from salvage therapy for advanced disease to potent agents used in the perioperative setting for curative intent therapy [5,6,7,8]. While some of the impact of ICI therapy results from targeting immunosuppressive molecules on tumor cells, in the majority of cases, the release of anti-tumor immune suppression comes from targeting immune cells in the TME. The clinical benefit from this therapy can be impressive in terms of progression-free and overall survival; however, long-term benefit from these therapies is found in less than 25% of all patients. These results immediately raise the possibility of developing therapies to target other cells and other molecules in the TME to build on this success. In addition, while initial studies were conducted in advanced-stage patients, some of the most important results providing biologic understanding of the clinical importance of the TME have come from these therapies tested in the neoadjuvant setting [9,10,11]. Importantly, tumor and TME specimens can generally be obtained both prior to therapy and after, from the surgical specimen, from which an immediate assessment of tumor response can be obtained. This assessment often includes the degree of major pathologic response, testing of predictive biomarkers of response, and comparison of the tumor and the TME pre- and immediately post-therapy [12]. Coordination between medical oncology and thoracic surgery teams is central to the conduct of these trials with respect to the timing of neoadjuvant therapy and resection and for harvesting of biospecimens to facilitate evaluation of predictive biomarkers and precision oncology efforts [9,13].

As such, there is intensive investigation into targeting other components of the NSCLC TME for potential therapeutic benefit. In this review, we provide an overview of the recent preclinical and clinical studies assessing the benefit of various therapeutic targets and cells in the NSCLC TME, reflect on the challenges of translating preclinical agents to the bedside, discuss the role of immunotherapy and other TME-specific agents in the neoadjuvant and adjuvant setting for early-stage resectable NSCLC, as well as identify the knowledge gaps we need to fill to allow progress. To provide a framework, in Figure 1, we identify multiple components of the TME and various therapeutic targets and therapies that are being evaluated for potential clinical benefit. As with most emerging novel therapies, most of these agents are under investigation in the translational stage, and those in clinical trials were first investigated in the setting of metastatic or advanced disease refractory to standard-of-care therapies. We hasten to add that it is already obvious that the TME has many more cellular components and/or subtypes of cells than those shown in Figure 1, but this Figure provides a starting point. In Table 1, we provide a working summary of current major knowledge gaps concerning the role of the TME in tumor biology and response to therapy, particularly immunotherapy.

## 2. Single-Cell Sequencing Analyses of the Tumor Microenvironment of NSCLC

The ecosystem of the NSCLC TME employs a dizzying array of crosstalk between tumor cells and surrounding TME components. To better understand therapeutic targets in the TME, it is first important to recognize that NSCLC tumors are highly heterogenous in their composition, with some more enriched for certain TME components compared to others. This heterogeneity in TME composition and its unique molecular signatures is a major knowledge gap that may directly impact how effective certain therapy types may be. Recent single-cell RNA-sequencing (scRNA-seq) characterization of this diversity has yielded important insight into both early and late-stage tumors in multiple cancer models, including NSCLC [14,15,16,17]. Wu et al. analyzed advanced NSCLC tumors from 42 patients and 90,406 cells and identified 11 major cell types, including cancer cells, non-cancer epithelial cells, immune cells, and stromal cells. Clustering analyses demonstrated aggregation of progenitor-type epithelial cells with their malignant counterparts as well as unique TME subtypes for several components, including endothelial cells, fibroblasts, tumor-associated macrophages (TAMs), regulatory T-cells (Tregs), dendritic cells (DC), and polymorphonuclear lymphocytes (PMNs) [15]. The authors demonstrated, through cell–cell interaction assays, several unique interactions between tumor cells and TME components through multiple signaling pathways. Indeed, they demonstrated that TAMs prominently suppressed T-cell function, which may be implicated in immune evasion [15]. Bischoff et al. similarly conducted scRNA-seq analyses from 10 lung cancer patients matched against 10 control patients and identified two major TME community patterns (CP^2^E: immune-activated and N^3^MC: immune-cold). CP^2^E-group tumors contained a greater proportion of cancer-associated myofibroblasts, pro-inflammatory macrophages, and exhausted CD8+ T-cells, while N^3^MC tumors comprised non-inflammatory macrophages, myeloid dendritic cells, normal myofibroblasts, and normal T-cells [14]. These stratifications had clinical significance; N^3^MC tumors were associated with superior overall survival (OS) compared to CP^2^E tumors [14].

Sc-RNAseq has yielded meaningful insights with respect not only to the heterogeneity of untreated tumors but also to how the TME changes in response to therapy. Maynard et al. demonstrated through a study of 49 samples from 30 patients that residual tumor cells after systemic therapy were enriched for an alveolar-regenerative state, while tumor cells from patients who developed progressive disease while receiving therapy were enriched for signatures associated with drug resistance [18]. As we strive to further precision oncology efforts to guide the best therapies for individual patients, the resolution of the TME components provided by Sc-RNAseq may be invaluable in identifying specific tumors that may be more or less sensitive to targeted therapies. The potential utility of Sc-RNAseq for these efforts reflects an ongoing knowledge gap.

## 3. Emerging Therapeutic Strategies Targeting the NSCLC TME

The TME of NSCLC is comprised of dense networks of stromal cells and proteins that comprise the extracellular matrix (ECM), cancer-associated fibroblasts (CAFs), and endothelial cells, as well as immune cells that directly contribute to tumor growth, invasiveness, and immune evasion [19] (Figure 1). At the same time, NSCLC cells themselves modulate TME cell activity through complex signaling networks and cross-talk with individual TME cellular players [20]. This complex network is also impacted by the presence of anti-tumor therapy, which induces responses in both tumor and TME cells. A prominent example of this is the secretion of pro-inflammatory cytokines by primary tumor cells that directly act on TME cells to augment tumor progression. Indeed, NSCLC cells secrete NF-kB, a key mediator of inflammation, which is correlated with poorer outcomes [21]. Inhibition of upstream and downstream targets of NF-kB are under investigation [22]. Similarly, CXCL1 expression is associated with recurrence after surgical resection and may be implicated in the recruitment of immunosuppressive Tregs into the NSCLC TME [23]. However, cytokine-specific inhibitors are challenging to implement clinically, as any given cytokine has innumerable functions in both benign and tumor cells, greatly limiting the ability to target pathologic pathways only. However, cytokine markers can be valuable clinical markers of therapy response [24].

Capturing these dynamic responses is a daunting task, as tumor cells themselves can be genetically distinct and heterogenous, with differential impacts on the surrounding TME. Perhaps the most clinically pertinent example of this is that NSCLC cells, alongside immune cells in the TME, express PD-L1 and contribute to immune evasion [25]. Additionally, tumor cell aneuploidy has been shown to be associated with reduced T-cell activation and worse OS [26]. The NSCLC histologic subtype profoundly impacts the TME and therapy response. Faruki et al. investigated immune cell genetic signatures from datasets encompassing 1190 adenocarcinoma (LUAD) cases and 761 squamous cell carcinoma (LUSC) cases. They identified complex, variable expression patterns of immune markers between both LC types and between subtypes for each cohort [27]. Several of these had clinical significance; higher expression of *CTLA4*, *CD274* (PD-L1), and *Major Histocompatibility Complex (MHC)*-II were associated with improved survival in the proximal inflammatory variants of LUAD, and higher expression of Th1, Th2, follicular T helper, dendritic cells, and MHC-II were associated with improved survival in primitive-subtype LUSC [27]. Furthermore, numerous studies have demonstrated that LUSC tumors have higher tumor mutational burden compared to LUAD tumors, which may increase their sensitivity to immunotherapy [28].

NSCLC driver mutational subtype has been demonstrated to directly impact TME behavior. Major driver mutations include *epidermal growth factor receptor* (EGFR), *Kristen rat sarcoma (KRAS), TP53, anaplastic lymphoma kinase (ALK), KEAP1, LKB1/STK11*, and others [29]. The two most common driver mutations are *KRAS* (29% of cases) and *EGFR* (19% of cases) [30]. *KRAS*-mutant tumors are more likely to have an immunosuppressive phenotype enriched with regulatory T-cells and M2-like TAMs, while *TP53*-mutant tumors are enriched with CD8-T-cells and PD-L1 positivity, making then potentially ideal targets for immunotherapy [26]. *ALK-*mutant NSCLC (5% of cases) that is resistant to targeted tyrosine kinase inhibitor (TKI) therapy has suppressed effector T-cell function and high levels of immunosuppressive regulatory T-cells in the TME [31]. Furthermore, higher expression of PD-L1 and CTLA-4 in these tumors portends poorer clinical outcomes after TKI therapy [32]. As we will discuss later, *EGFR-*mutant NSCLC is notoriously resistant to immunotherapy, which may be due in part to its uniquely immunosuppressive TME and low tumor mutational burden [33].

Environmental factors also impact the TME and therapy response. Smoking is a well-established inducer of somatic mutations and neoantigen production in tumor cells, which has been shown in whole-exome sequencing studies and a meta-analysis of six trials to enhance the efficacy of immunotherapy [34,35]. However, another recent meta-analysis of 17 phase III trials demonstrated no significant impact of smoking status on immunotherapy [36]. Continued investigation into this dense cross-signaling network is an ongoing knowledge gap that is under intensive investigation, which may shed light on special patient populations or tumor oncogenotypes that may be more or less responsive to TME-specific targeted therapies.

### 3.1. Structural Components

#### 3.1.1. Endothelial Cells

Endothelial cells (ECs) form the innermost single layer of blood and lymphatic vessels and are the primary mediators of angiogenesis, a fundamental hallmark of cancer progression with respect to maintenance of perfusion to the primary tumor and to pave potential pathways for metastatic spread [3]. Angiogenesis is facilitated by hypoxia in the TME through multiple cytokines and growth factors that are secreted by tumor cells and other TME cellular constituents, including TAMs, PMNs, and cancer-associated fibroblasts [37,38,39]. These factors include hypoxia-inducible factor-1α (HIF-1α), epidermal growth factor (EGF), fibroblast growth factor-2 (FGF-2), and vascular endothelial growth factor (VEGF), among others. Each of these factors has been directly implicated in tumor angiogenesis in the TME and tumor growth, epithelial-mesenchymal transition (EMT), invasiveness, and remodeling of the extracellular matrix (ECM), as well as chemoresistance. To this point, ECs exhibit considerate phenotypic plasticity, enabling a transition towards a mesenchymal phenotype (termed the endothelial–mesenchymal transition). This complex process progresses alongside EMT and contributes to therapy resistance and metastasis [40,41].

Abrogation of tumor angiogenesis is a well-established strategy in NSCLC. Bevacizumab (anti-VEGF monoclonal antibody) is Food and Drug Administration (FDA)-approved as a first-line therapy in select patients, often in combination with carboplatin and paclitaxel [42,43,44]. Furthermore, ramucirumab is approved as a second-line agent [45]. There may be a promising therapeutic interface between *EGFR-*mutant NSCLC and ECs in the TME. While a small phase II trial investigating combination therapy with gefitinib (EGFR inhibitor) + bevacizumab for *EGFR-*mutant NSCLC demonstrated inferior PFS compared to gefitinib alone [46], Saito et al. demonstrated that bevacizumab + erlotinib improved PFS on interim analysis compared to erlotinib alone [47]. These mixed findings may be due, in part, to the mechanism of drug delivery into the TME. Zhao et al. designed a novel nanoparticle-loading platform for bevacizumab and gefitinib that releases bevacizumab in the TME with oxidation and then gefitinib once the apparatus is internalized intracellularly. This dual-therapy platform suppressed NSCLC tumor growth in vivo [48]. Anti-angiogenic combination therapy in NSCLC has evolved into the immunotherapy realm as well. It is well known that aberrant tumor angiogenesis, mediated by VEGF, is implicated in the polarization of TAMs, recruitment of immunosuppressive regulatory T-cells (Tregs), and suppression of CD8+ T-cell function [49]. Concordantly, the abrogation of aberrant tumor angiogenesis may suppress immune evasion and potentiate anti-tumor immune responses. This has been recently investigated clinically in a phase II, non-randomized study of 38 treatment-naive patients with advanced nonsquamous NSCLC, no *EGFR, STIK11, MDM2,* or *ROS1* mutations, and a tumor mutational burden (TMB) > 10 mutations/megabase who received bevacizumab + atezolizumab. Combination therapy yielded a 72% 1-year OS, 42.1% ORR, and 78.9% disease control rate (percentage of patients who achieved partial or complete response or with stable disease) [50]. Furthermore, Sugawara et al. similarly demonstrated that combination therapy with nivolumab + paclitaxel/carboplatin/bevacizumab was superior to placebo + paclitaxel/carboplatin/bevacizumab for untreated, advanced nonsquamous NSCLC (61.5 vs. 50.5% ORR, respectively) [51].

#### 3.1.2. Cancer-Associated Fibroblasts

Cancer-associated fibroblasts (CAFs) maintain and dynamically remodel the ECM in the NSCLC TME and are implicated in metastasis, EMT, angiogenesis, chemoresistance, and immunosuppression [52,53,54]. CAFs produce and modify ECM components, including collagen, fibronectin, and glycosaminoglycans, which can alter the physical properties of the tumor stroma and influence tumor cell behavior [55]. They also modulate the antitumor immune response by promoting recruitment of regulatory T-cells and suppressing cytotoxic T-cell function [56,57]. Like other TME cells, NSCLC CAFs are heterogenous in their cellular signatures and functions but can be globally characterized into inflammatory, myofibroblastic, or antigen-presenting subtypes [53,58,59]. In the context of NSCLC, recent cytometric analyses demonstrate that inflammatory subtypes may be associated with better outcomes, while other matrix-associated subtypes may be associated with worse outcomes [60]. Furthermore, CAF populations are dynamic with respect to therapy status, which may suggest ECM remodeling as a response to treatment [60].

As such, suppressing the pro-tumorigenic function of CAFs is a promising therapeutic strategy. In the preclinical setting, fibroblast-activating protein (FAP)+ CAFs were found to be associated with tumor progression, and inhibition of FAP activity resulted in antitumor responses [61]. FAP inhibition in NSCLC has yet to yield durable clinical benefit [62]. Similarly, anti-fibroblast growth factor (FGF)-R therapy with rogaratinib also did not improve PFS in a phase II study of high-FGFR1-3 mRNA-expressing squamous cell NSCLC (LUSC) [63]. Interestingly, Gao et al. presented a case report of two patients in whom clinical benefit was noted with the FGFR inhibitor pemigatinib in FGFR-mutant NSCLC [64]. Continued investigation into the FGF/FGFR axis inhibition in NSCLC is warranted.

IL-6 may be a promising therapeutic target. Prior studies have demonstrated IL-6 secretions from CAFs promote EMT and chemoresistance [62]. Inhibition of IL-6 is currently being investigated in clinical trials. Recently. Du et al. presented a single-center study that an IL-6 inhibitor, tocilizumab, improved OS, as well as several serum markers of nutrition and inflammation, in patients with advanced NSCLC and cachexia compared to conventional antitumor therapy [65]. Furthermore, multiple combination tocilizumab + immunotherapy trials are in process (NCT04691817, NCT06016179).

Transforming growth factor (TGF)-β is the master regulatory cytokine for CAF activity in the NSCLC TME and is directly associated with EMT, stemness, and tumor progression [66,67]. Activation of TGF-β and its downstream Smad signaling pathway is implicated in solid-to-acinar transition in LUAD [68]. Inhibition of TGF-β or its downstream targets is a complex and evolving issue in NSCLC. In other cancers, direct inhibition of TGF-β in clinical trials has yielded mixed results [69,70,71]. In NSCLC, a phase II trial of bifunctional PD-L1/TGF-β inhibitor, SHR1701, is in process (NCT04560244).

### 3.2. Immune Components

#### 3.2.1. Neutrophils

Tumor-associated neutrophils (TANs) are a functionally diverse cohort of immune cells that are highly abundant in the NSCLC TME. They are more frequently identified in squamous cell carcinoma compared to adenocarcinoma [72]. TANs contribute to multiple aspects of cancer progression, including global immunosuppression, formation of neutrophil extracellular traps (NETs), maintenance of chronic inflammation, angiogenesis, EMT, and metastasis [72,73,74,75,76]. However, subsets of polarized TANs also stimulate antitumor T-cell response and damage tumor cells directly [39,72,77]. Single-cell RNA sequencing analyses have evolved our understanding of TAN polarization from the archetypal N1 (anti-tumorigenic) and N2 (pro-tumorigenic) phenotypes to a heterogenous and highly plastic cohort of cells, which may be pro-angiogenic, immunomodulatory, hybrid, interferon-responsive, or matured/primed [72]. Knowledge of such phenotypic heterogeneity suggests fascinating avenues for investigation into repolarization tactics or inhibition of immunosuppressive functions and, at the same time, highlights the inherent limitations of a “pan-TAN” inhibitor therapeutic approach. It is clear that holistically, TAN presence carries prognostic significance, and an elevated serum neutrophil count and high neutrophil/lymphocyte ratio is associated with poorer prognosis and diminished therapy response rate [73,78].

A promising neutrophil-specific therapeutic strategy pertains to the use of nanoparticles for drug delivery. In a colorectal cancer murine model, Chen et al. demonstrated that DNase 1 released by a photoregulated nanoparticle was effective in deconstructing NETs and increasing tumor immunosensitivity [79]. Another direction is using inhaled empty virus nanoparticles. Additionally, abrogation of TAN recruitment through inhibition of CXCR2 may augment the antitumor efficacy of cisplatin. CXCR2 expression is associated with TAN recruitment into the TME, EMT, and poor prognosis in NSCLC [80]. Cheng et al. demonstrated that SB225002, a CXCR2 inhibitor, not only suppressed NSCLC tumor cell EMT in vitro and tumor growth in vivo but also decreased N2-TAN infiltration in the TME [80]. Furthermore, combination therapy with SB225002 and cisplatin was synergistically more effective in reducing tumor growth in an orthotopic murine model, and CXCR2 monotherapy prevented enhanced CD8+ T-cell activation [80]. These findings highlight the continued importance of exploring neutrophil-specific agents as part of a multimodal strategy for NSCLC.

#### 3.2.2. Regulatory T-Cells

FoxP3+/CD4+/CD25+ regulatory T-cells (Tregs) are intimately involved in the maintenance of immune homeostasis and immunosuppression [81]. In NSCLC, the presence of Tregs in the TME is associated with metastasis, immune evasion, and poorer prognosis [82,83,84]. Tregs accumulate in the TME in response to chemokines such as CCL22, CCL17, and CXCL12, which are secreted by both tumor cells and stromal components, including CAFs [85]. Here, Tregs exert suppressive effects by inhibiting cytotoxic CD8+ T-cells and natural killer cells through the production of immunosuppressive cytokines such as IL-10, TGF-β, and IL-35 and by direct cell–cell contact via molecules such as CTLA-4 [86]. Tregs also impair dendritic cell maturation and antigen presentation, further blunting antitumor immune response. Their suppressive dominance within the TME contributes not only to immune tolerance but also to resistance against immunotherapies, including checkpoint blockade strategies targeting PD-1/PD-L1 or CTLA-4 [87]. Even for early-stage disease, Treg presence is associated with recurrence after resection, particularly in node-negative disease [87,88].

While the suppression of Treg activity is promising, there have been relatively few identified agents that specifically abrogate Treg activity with meaningful signal in the preclinical setting. This is a clear knowledge gap with respect to targeting an extremely important immunosuppressive cellular player in the NSCLC TME. Song et al. recently identified two anti-CD25 antibodies (BA9 and BT942) that exhibited cytotoxic activity in vivo and suppressed tumor growth in a colorectal cancer humanized mouse model both as a monotherapy and in combination with anti-PD-1 therapy [89]. More investigation into the therapeutic targeting of Treg activity in NSCLC is necessary.

#### 3.2.3. Dendritic Cells

Dendritic cells (DCs) function to stimulate CD8+ and CD4+ T-cell responses through the presentation of tumor-associated antigens. It has been observed that impaired antigen presentation results in resistance to immune checkpoint inhibitor therapy [90]. Moreover, the presence of intratumoral DC populations is associated with improved survival [91]. Thus, DC-based therapeutic strategies have been explored as an option to stimulate the anti-tumor immune responses and potentiate the efficacy of immune checkpoint blockade.

DCs are potent antigen-presenting cells (APC) and thus represent a potential strategy to improve the efficacy of tumor vaccines. The theory behind this approach is that direct intratumoral administration of DCs, or the administration of neoantigens and growth factors that induce DC proliferation, will generate a population of APCs with direct access to an abundance of neoantigens. In a mouse model of NSCLC, Lopez et al. delivered Flt3 to expand DC populations and αCD40 to activate them. This approach induced a cytotoxic T-cell response and amplified the host response to weak neoantigens [92]. Similarly, Lim et al. performed intratumoral injection of DCs into an NSCLC murine model and observed higher intertumoral T-cell infiltration and potentiation of the efficacy of anti-PD1 therapy [93]. This approach has recently been evaluated in clinical trials. In the first human trial utilizing a neoantigen-targeted DC vaccine as adjuvant therapy for NSCLC patients, Ingels et al. observed durable and diverse T-cell populations in treated patients, with good tolerance of therapy [94]. Similarly, Vounckx et al. performed a phase II clinical trial evaluating the combination therapy approach of anti-PD1 therapy, stereotactic body radiation therapy, and intratumoral myeloid DC injections in multiple solid tumors, including eight NSCLC patients. While the trial did not reach its pre-specified endpoint of improved OS, a subset of patients experienced a radiographic response [95].

DCs have also been used in adoptive cell transfer protocols. Chen et al. report the results of a phase I trial, which included five NSCLC patients, in which patients received a single infusion of DCs and cytokine-induced natural killer cells. The authors found that the treatment was safe and feasible [96]. Additionally, a phase III clinical trial evaluated adjuvant adoptive transfer of activated cytotoxic T-cells and DCs as a potential therapeutic strategy in the adjuvant setting. Patients were randomized to receive either immunotherapy, which included the adoptive transfer of autologous activated killer T-cells and dendritic cells along with chemotherapy versus standard chemotherapy alone. Five-year overall survival rates were 69% in the adoptive cell transfer arm compared to only 45% in the chemotherapy arm [97].

#### 3.2.4. Tumor-Associated Macrophages

Macrophages, like TANs, are abundant in the NSCLC TME. Phenotypically, tumor-associated macrophages (TAMs) were classically stratified into a pro-inflammatory, antitumorigenic M1 state and a pro-tumorigenic, immunosuppressive M2 state [98]. In reality, TAMs exist on a spectrum between these states, and their genetic signatures are modulated by tumor progression and exposure to therapy [18]. This is a major knowledge gap in our understanding of macrophage biology in NSCLC and its clinical impact with respect to sensitivity to immunotherapy and immune evasion. Prior investigations from our group have demonstrated that NSCLC cells strongly induce the high expression of immunosuppressive *Arginase-1* in co-cultured murine macrophages in vitro and in vivo and polarize human macrophages to an M2-like state in co-culture studies [99]. TAMs are implicated in every aspect of tumor progression, including angiogenesis, extracellular matrix remodeling, metastasis, immune evasion, and chemotherapy resistance [37,98,100].

Due to their prominent role in NSCLC progression, efforts to target macrophages to suppress their immunosuppressive effects or repolarize M2-like macrophages to an M1-like state are under intensive investigation. A promising TAM-specific strategy is to suppress M2-like activity to potentiate the effect of anti-PD1 immunotherapy. Recently, Zhang et al. demonstrated that pexidartinib, a C-Kit and macrophage-colony stimulating factor inhibitor, reduced TAM expression of CCL22, a Treg chemoattractant, in NSCLC cells in vitro; combination therapy with anti-PD-1 therapy synergistically improved survival and reduced tumor weight in a murine model and significantly increased the CD8+/Treg T-cell ratio [101].

Restricting M2-like polarization in NSCLC TAMs may also be a promising therapeutic strategy. A well-characterized receptor implicated in M2 polarization is the adenosine A_2A_ receptor (A2AR), which binds adenosine produced from ATP metabolism. Lei et al. recently demonstrated that NSCLC cells and TAMs secreted adenosine, which promoted A2AR activation in TAMs. A2AR activation promoted the upregulation and secretion of CXCL5 via increased expression of NF-kB. This resulted in the augmentation of NETosis by TANs and suppression of CD8+ T-cell activity [102]. Promisingly, A2AR inhibition by the small molecule CPI-444 and CXCR2 inhibition by the small molecule SB225002 abrogated tumor growth and NET area in murine xenografts; CPI-444 treatment suppressed CXCL5 expression [102]. A clinical trial investigating CPI-444 alongside a variety of other combination immunotherapeutic agents for metastatic NSCLC is underway (NCT03337698).

NSCLC cells secrete monocyte chemoattractant protein-1 (also known as chemokine C-C motif ligand-2 (CCL2)), thereby recruiting macrophages to the TME and appearing to polarize them to an M2-like state [103]. While MCP-1 presence was associated with superior survival in clinical samples, suppression of CCR2 (CCL2 receptor) suppressed A549 cell growth and migration via inhibition of matrix metalloproteinase 9 (MMP9) [103,104]. Recently, Chen et al. demonstrated that CCL2 inhibition by the upstream silencing of the PIM1 proto-oncogene in NSCLC cells abrogated TAM polarization to an M2-like state and recruitment to the TME. PIM1 was identified in nearly 40% of clinical NSCLC samples and was significantly associated with smoking status and inferior overall survival [105]. Furthermore, PIM1 inhibition augmented the antitumor activity of PD-1 inhibitors in murine models [105].

NSCLC cells augment immune evasion by suppressing phagocytic activity by TAMs. CD47 is an integrin-associated protein that is highly expressed by NSCLC cells and is associated with more aggressive disease biology and poor prognosis [106]. CD47 modulates numerous downstream pathways, including binding of cell surface receptor SIRPα, suppressing phagocytotic capacity in macrophages [106]. Preclinical studies in other tumor models have demonstrated that CD47 blockade permits tumor-cell phagocytosis by macrophages with subsequent priming of CD8+ T-cells and suppression of immunosuppressive Tregs [107]. Furthermore, CD47 inhibition may enhance the anti-tumor activity of cisplatin against NSCLC cells and further augments macrophage-mediated phagocytosis [108]. As such, several early and middle-stage clinical trials investigating anti-CD47/SIRPα therapies are in process for NSCLC and other malignancies [106].

#### 3.2.5. CD8+ T-Cells and the Immunotherapy Revolution

CD8+ T-cells are the primary mediators of anti-tumor immune response in NSCLC. Arguably the most important advancement in the treatment of cancer of the 21st century, thus far, is the advent of immunotherapies that augment CD8+ T-cell response by inhibiting the PD-1/PD-L1 axis. Indeed, the 2018 Nobel Prize in Medicine was awarded for the discovery of immune checkpoint inhibitor therapy [109]. First utilized for the treatment of advanced melanoma, anti-PD-1/PD-L1 and anti-CTLA-4 therapies have revolutionized the management of advanced NSCLC, with multiple agents that are now FDA-approved for various disease stages, including pembrolizumab [110], atezolizumab [111], nivolumab [112], cemiplimab [113], ipilimumab [114], tremelimumab [115], and durvalumab [116,117].

ICI therapy is based on two major mechanisms through which tumor immune evasion capabilities are suppressed and anti-tumor activity by CD8+ T-cells is potentiated: inhibition of cytotoxic T-lymphocyte associated protein 4 (CTLA4) and inhibition of the programmed death 1 (PD-1) or PD ligand 1 (PD-L1) axis (Figure 2).

CTLA-4 competitively inhibits the binding of CD28 to B7 on antigen-presenting cells, which prevents the co-stimulatory signal activation of CD8+ T-cells [24,118]. CTLA-4 is expressed by Tregs, other T-cells, and even by tumor cells in the TME [119,120]. In NSCLC, Paulsen et al. demonstrated that tumor-cell expressed CTLA-4 in metastatic lymph nodes was associated with poorer survival, but stromal CTLA-4 expression was associated with improved LUSC disease-specific survival [121]. This equivocal prognostic impact in tissue specimens correlated with a mixed clinical response to ipilimumab in combination with chemotherapy. In a trial of 749 patients with advanced, chemotherapy-naive LUSC, Govindan et al. demonstrated that ipilimumab + chemotherapy did not significantly improve OS compared to chemotherapy alone (though dose toxicity and the inability to tolerate treatment may have modulated outcomes) [122]. These findings bring into question the efficacy of anti-CTLA-4 monotherapy for ICI in NSCLC; however, dual ICI in combination with PD-1/PD-L1 blockade may be a promising strategy [123]. Indeed, combination therapy with ipilimumab + nivolumab for recurrent or stage IV disease has demonstrated improved OS compared to chemotherapy alone, regardless of PD-L1 positivity level [124].

Both NSCLC cells as well as their TME components express PD-L1, including TAMs [125] and CAFs [126]. Upon the binding of PD-L1 to PD-1 on CD8+ T-cells, downstream activation of CD8+ T-cells, their proliferation, and their functional activity are suppressed [127]. Up to 60% of NSCLC tumors express PD-L1, and positivity is classically defined as expression in 5% or more of NSCLC cells in a sample [25]. Therapeutic inhibition of the PD-1/PD-L1 cascade has yielded substantial improvement in cancer-specific survival in NSCLC patients. Since 2015, numerous studies have demonstrated durable clinical benefit with PD-1 inhibition, either as monotherapy or in combination with chemotherapy, for advanced NSCLC [128,129,130,131,132,133,134,135].

#### 3.2.6. Building upon the Immunotherapy Foundation: Factors and Biomarkers Predicting Response to ICI Therapy

Despite these achievements, only up to 30-50% of patients derive clinical benefit from ICI [136,137,138]. Factors modulating ICI efficacy are under intense investigation and are likely multifactorial, reflecting a major knowledge gap. A recent investigation by Patel et al. suggests that males may have a superior response to chemoimmunotherapy compared to females, while age, race, and Charleson comorbidity index do not impact efficacy [139]. Smoking status is a well-established marker for superior response to ICI [140]. A logical biomarker for ICI responsiveness is PD-L1 positivity status, and indeed, first-line single-agent atezolizumab or pembrolizumab is utilized with tumor proportion score (TPS) ≥ 50% and as second-line therapy with TPS ≥ 1% [141]. However, PD-L1 assays are inconsistent, and in other tumor models, TPS (which assays for PD-L1 expression on tumor cells) has not been shown to correlate with ICI responsiveness [141]. Even in NSCLC, a meta-analysis of multiple RCTs demonstrated that the ICI benefit as second-line therapy persists independently of PD-L1 status [142]. Combined Positive Score (CPS), which accounts for PD-L1 expression in tumor and TME cells, may be a superior predictor of clinical outcomes after ICI [143]. Prior smoking status may also correlate with ICI responsiveness. A meta-analysis of 4971 patients demonstrated a survival benefit with ICI vs. chemotherapy in current or former smokers but not never-smokers [142]. This may be due, in part, to the higher prevalence of *EGFR* or *ALK* mutations in this cohort, as these patients are less likely to respond to ICI [142].

Another well-established biomarker for ICI responsiveness is tumor mutational burden (TMB), which refers to the number of somatic mutations in the tumor genome. The higher the TMB, the greater the number of tumor-associated neoantigens that may increase T-cell responsiveness and antitumor immune response [144,145,146]. In a series of 1552 NSCLC patients, Ricciuti et al. demonstrated that higher TMB was significantly associated with improved therapy response, CD8+ T-cell infiltration, and survival after PD-1/PD-L1 inhibitor therapy [147]. The standardization of TMB assays and the study of other factors that affect TMB and recognition of neoantigens by CD8+ T-cells (i.e., microsatellite instability status, exogenous mutants, and others) warrant further investigation [148].

Oncogenotype also impacts ICI responsiveness. Meta-analyses have demonstrated that *EGFR-*mutant NSCLC is less sensitive to ICI, which may be due, in part, to a more immunosuppressive TME in these tumors [138,145,149]. Indeed, *EGFR*-mutant NSCLC tends to have lower TMB and variable expression of PD-L1, which may contribute to ICI resistance [33,150]. *EGFR*-mutant tumors are also enriched with Tregs and M2-TAMs and have a low presence of CD8+ T-cells, which further increases immune evasion [33]. Despite the degree of immunosuppression in the TME, *EGFR*-mutant tumors generally have a better prognosis compared to wild-type tumors, and several generations of EGFR-specific tyrosine kinase inhibitors have been approved, with durable clinical benefit [151]. Unfortunately, therapy resistance remains a crucial problem, which may be potentiated by the immunosuppressive microenvironment [33]. Efforts to combine EGFR-TKI with ICI have shown mixed results. Early studies showed no significant improvement in ORR compared to monotherapy and significant concerns regarding toxicity were raised [152,153]. As such, targeting key downstream targets of the *EGFR* signaling cascade (i.e., CD73, an enzyme which dephosphorylates adenosine monophosphate into adenosine, which suppresses T-cell function) alongside ICI may be a promising strategy and is under investigation in the preclinical setting [154,155]. Furthermore, while *KRAS-*mutant tumors are particularly more sensitive to ICI, the presence of *LKB1/STK11* mutations in conjunction with *KRAS-*mutant disease is also associated with increased ICI resistance [138,145,156]. Recruitment of TANs into the TME through secretion of pro-inflammatory cytokines may be implicated in ICI resistance in this tumor subtype [157]. The molecular mechanisms underlying ICI resistance in these mutational subtypes overall is a key knowledge gap.

A major aspect of ICI resistance lies in the contributing roles of other TME cells that may promote immunotherapy resistance, and several of the TME-specific agents previously discussed have been investigated in combination with ICI to enhance immunosensitivity. Certainly, tumor cells themselves upregulate resistance pathways that promote ICI resistance [138]. Furthermore, the presence and phenotype of T-cells in the NSCLC TME are modulated by presence and response to therapy. Indeed, scRNA-sequencing analyses demonstrate that tumor samples from residual disease after therapy are enriched for a more inflamed TME with a greater presence of T-cells, which may be a potentially clinically relevant opportunity to initiate immunotherapy if not already started [18]. This is further supported by a meta-analysis of 1833 patients that demonstrated that a higher density of infiltrating CD8+ T-cells was associated with improved survival after ICI for metastatic disease [158]. Wu et al. recently demonstrated that M2-TAMs were enriched in nivolumab-resistant clinical NSCLC specimens, and in vivo, upregulated expression of *METTL3* and m6A RNA may be implicated in ICI resistance [159]. Palliyage et al. demonstrated that CAFs increased the expression of PD-L1 as a response to chemotherapy and potentiated stemness and invasiveness in a PD-L1 dependent manner through the increased expression of hepatocyte growth factor [126]. Curiously, while Treg infiltration in NSCLC is associated with further immunosuppression in the TME and poorer prognosis, it may be associated with more ICI sensitivity [81]. This paradoxical effect may be due, in part, to the presence of PD-L1+ Tregs that could, conceivably, increase overall PD-L1 positivity and therefore increase sensitivity to anti-PD-L1 therapy [81].

The future of immunotherapy in NSCLC, therefore, rests on increasing the number of patients who may benefit from its effects, either by combining ICI with other receptor or TME-specific targeted therapies beyond chemotherapy or by utilizing immunotherapy in the perioperative setting to enhance outcomes for patients with early-stage disease undergoing surgical resection with curative intent.

Combination therapy is an area of active investigation and a key knowledge gap. A recent Bayesian network meta-analysis of 8647 patients in 14 RCTs demonstrated that combination ICI + anti-angiogenic therapy (bevacizumab) + chemotherapy was associated with superior PFS and ORR in PD-L1-high patients [160]. Furthermore, multiple ongoing trials are investigating dual-ICI therapies, ICI + TKI (e.g., axitinib, ramucirumab, lenvatinib, and others), as well as cancer vaccinations [161]. We note that several TKIs under investigation impact several tyrosine kinase pathways, including angiogenesis. This may modulate their efficacy in combination therapy strategies. As an example, combination therapy with pembrolizumab and lenvatinib for PD-L1+, treatment-naive metastatic NSCLC did not improve mOS in a recent phase III trial of 623 patients [162]. Recently, Besse et al. conducted a phase II umbrella study specifically investigating biomarker-specific combination therapy with durvalumab for advanced NSCLC. A total of 286 patients were included, and combination therapy with ceralsertib (ataxia telangiectasia and Rad3-related (ATR) kinase inhibitor) demonstrated superior ORR compared to combination therapies with poly-(ADP ribose) polymerase (PARP) inhibitors, STAT3 antisense oligonucleotide, or anti-CD-73 antibody therapy [163]. Overall, combination immunotherapy with TME-specific targeted therapies is under investigation, though with mixed results. However, as reflected by Desai et al., identifying biomarkers that could potentially guide which patient and tumor characteristics could best lend themselves to a specific regimen remains a key priority [161].

A particularly promising area of interest is combination therapy with ICI and radiation or other energy-delivery therapies. The “abscopal effect” of radiation therapy refers to the stimulation of the local TME and production of neoantigens after treatment that suppresses further tumor progression and metastasis through immune activation. As such, combining ICI with radiation is a potent strategy to reinforce antitumor immune response and potentially reduce radiation resistance [164]. Indeed, secondary analysis of a single-institution subset of patients from the original KEYNOTE-001 study demonstrated improved OS in radiated patients treated with pembrolizumab compared to those with no prior radiotherapy [165]. The phase III PACIFIC trial further validated this principle and demonstrated that consolidation ICI improved PFS after definitive chemoradiation for stage III NSCLC compared to placebo [132]. In clinical specimens from patients treated with radiation + pembrolizumab, van der Woude et al. demonstrated that combination therapy increased cytotoxic T-cell infiltration significantly more than ICI alone [166]. The interplay between other TME components and radiation exposure is an evolving discipline that may shed light into radiotherapy resistance and enhancing antitumor immunity after combination therapy. As an example, Yang et al. recently demonstrated in a murine NSCLC model that radiation therapy induced increased infiltration of Ccl8+ M2-TAMs and elevated expression of immunosuppressive genes, suggesting a possible reactive mechanism of therapy resistance [167].

## 4. Perioperative Immunotherapy and TME-Specific Implications for the Thoracic Surgeon

Until recently, ICI was typically reserved for patients with advanced, unresectable disease. However, given its significant clinical benefit in this setting, increasing efforts are being made to implement ICI in the perioperative setting as either neoadjuvant or adjuvant therapy for resectable or borderline-resectable disease. This paradigm shift carries significant implications for the thoracic surgeon. Currently, up to stage IIIA disease (T3N1, in select cases) is potentially resectable. Conversely, bulky N2 disease and most cases of stage IIIB or stage IV disease are considered unresectable [168]. Unfortunately for patients with higher-stage resectable disease undergoing multimodality treatment, including surgery, OS rates are still suboptimal and estimated to be between 13 and 36% due to recurrent local or metastatic disease [169]. Furthermore, the established dogma of surgery + adjuvant platinum-based chemotherapy has only improved 5-year survival by 5.4% in pooled analyses [170]. As such, there is a pressing imperative both to increase the number of patients who could potentially be downstaged to become surgical candidates and to suppress local recurrence or micrometastatic disease spread after potentially curative resection and improve long-term outcomes.

A putative advantage of neoadjuvant ICI would be to decrease tumor size and increase the probability of achieving an R0 resection. The biologic mechanisms underlying the benefits of perioperative immunotherapy continue to be elucidated. In a murine model of NSCLC, Cascone et al. demonstrated that neoadjuvant anti-PD-1, anti-CTLA-4, or combination ICI followed by resection 2 days after therapy resulted in improved survival compared to adjuvant therapy. Furthermore, there was a significantly greater density of tumor-infiltrating lymphocytes and CD8+ T-cells in the TME in the neoadjuvant arm [171].

The landmark CheckMate 816 trial demonstrated that neoadjuvant nivolumab + platinum chemotherapy resulted in longer event-free survival and improved pathologic complete response rates without compromising surgical feasibility or increasing the adverse event rate [10]. Similarly, Heymach et al. and Wakelee et al. recently demonstrated in two phase III trials that perioperative (neoadjuvant and adjuvant) durvalumab [11] and pembrolizumab [172] + platinum-based neoadjuvant chemotherapy significantly improved PRR and event-free survival. Combination therapy has also yielded promising clinical results, as well as evidence of increased immune cell infiltration in the TME. Building upon their preclinical findings discussed above, Cascone et al. reported in the phase II NEOSTAR trial that neoadjuvant ipilimumab + nivolumab yielded a higher pathologic complete response rate vs. nivolumab alone (38 vs. 10%) and a higher frequency of memory T-cells in tumor specimens [9]. Furthermore, neoadjuvant ICI monotherapy has also been demonstrated to provide clinical benefit; Rusch et al. demonstrated that neoadjuvant atezolizumab yielded a major pathologic response rate in 20% of patients and a pathologic complete response in 6% of patients. Importantly, there were clinically insignificant effects of neoadjuvant ICI on preoperative pulmonary function tests and no significant differences in the rate of perioperative complications [173].

In the adjuvant setting, the Impower010 trial of 1269 patients with resected NSCLC demonstrated that those patients randomized to adjuvant atezolizumab vs. best supportive care had superior DFS, particularly patients with PD-L1% > 1% [174]. Similarly, the KEYNOTE-091 trial of 1177 patients with resected stage IB-IIIA NSCLC showed improved DFS compared to placebo, regardless of PD-L1 status [175].

Given these promising findings of the clinical benefit of ICI for even early-stage patients, atezolizumab, nivolumab, pembrolizumab, and, most recently, durvalumab have been FDA-approved in the perioperative setting for resectable NSCLC [176,177,178,179] (Figure 2). Taken collectively, a large meta-analysis of 43 studies and 5431 patients who underwent neoadjuvant chemoimmunotherapy demonstrated that even in patients with tumor PD-L1 < 1%, there was a significant improvement in event-free survival (but not OS) compared with chemotherapy alone [180]. Whether ICI is favored in the neoadjuvant or adjuvant settings is a knowledge gap that requires further exploration.

The next forays of investigation into harnessing the full potential of TME-specific targets in this patient population include combining neoadjuvant ICI and other immune-specific targeted therapies [181]. As an example, Cords et al. recently demonstrated that tumors from patients who received resection and adjuvant therapy who did not suffer from relapse were enriched with IDO (indoleamine dioxygenase) + (interferon-response) CAFs, while those patients with relapse after surgery had tumors enriched with pro-tumorigenic, immunosuppressive, and hypoxia-associated CAFs [60]. This accords with the notion that an inflamed TME may be more ICI-responsive in the perioperative setting. Furthermore, intratumoral immunotherapy is under active investigation. Predina et al. demonstrated in a murine model that intratumoral immunotherapy was highly effective in reducing tumor volume with enhanced CD8+ T-cell antitumor activity in early stage disease and less so in advanced disease [182]. These promising findings may have important implications for either preoperative immunostimulation or intraoperative intratumoral injections to enhance the efficacy of adjuvant immunotherapy. Overall, it is clear that the immune response to neoadjuvant ICI carries significant clinical impact. Laza-Briviesca et al. studied serum immune-related changes associated with complete pathologic response between the time of diagnosis and post-ICI. Patients with complete pathologic response had higher levels of PD-1+ cells, NKG2D+ CD56 T-cells, and CD4 T-cells; lower levels of CTLA-4+ in NK cells and monocytes; and numerous phenotypic changes compared to patients who did not sustain a complete pathologic response [12].

Given the concerns for autoimmune pneumonitis and interstitial lung disease after ICI, there is concern that these events may be associated with tissue friability and increased intrathoracic fibrotic burden. As such, this may increase the technical complexity of surgical resection with the potentially increased risk of conversion to open thoracotomy from a planned minimally invasive approach, intraoperative bleeding, or incomplete resection [183]. While recent investigations have demonstrated no significant differences in the capacity to complete safe surgical resection, it is clear that neoadjuvant ICI can significantly modulate surgical decision-making [184]. Bott et al. demonstrated in a phase I study of 20 patients who received neoadjuvant nivolumab that 54% of patients who were planned for video-assisted thoracoscopic surgery required conversion to open thoracotomy due to hilar inflammation and fibrosis [185]. Treatment of ICI toxicity is based on severity; low-grade disease may be addressed with temporary therapy interruption, and more severe disease may require steroids or anti-inflammatory biologic therapy [186]. Indeed, the TME itself may be a therapeutic target in this regard as well, though this is a major current knowledge gap. As an example, disulfiram, a FROUNT (regulator of CCR2/CCR5 signaling cascade) inhibitor, has been shown to suppress the pro-tumorigenic activity of TAMs in NSCLC [187]. It has also been recently investigated as a potent inhibitor of macrophage infiltration in bleomycin-induced fibrosis in preclinical studies [188]. Further exploration into the role of such anti-fibrotic agents to ameliorate ICI-mediated fibrosis is necessary. Overall, as neoadjuvant ICI continues to be implemented as the standard of care for resectable NSCLC, its impact on surgical feasibility and safety warrants continued investigation.

## 5. Conclusions

The NSCLC tumor microenvironment remains a dense, yet fertile soil that directly contributes to each hallmark of cancer progression. As the advent of immunotherapy continues to revolutionize the care of patients suffering from NSCLC, the fervency of investigation into additional potential therapeutic targets in the TME (often in combination with immune checkpoint blockade therapy) is palpable. This enthusiasm has even extended from advanced, unresectable disease to the ever-expanding role of immunotherapy to improve outcomes in early-stage disease in the perioperative period. As the landscape for TME-specific therapies for non-small cell lung cancer continues to expand, it will be imperative for the translational and clinical oncology community to remain abreast not only of opportunities to improve oncologic outcomes but also of new TME-specific adverse effects that may impact therapy utility and safety.

## Figures and Tables

**Figure 1 cancers-17-01732-f001:**
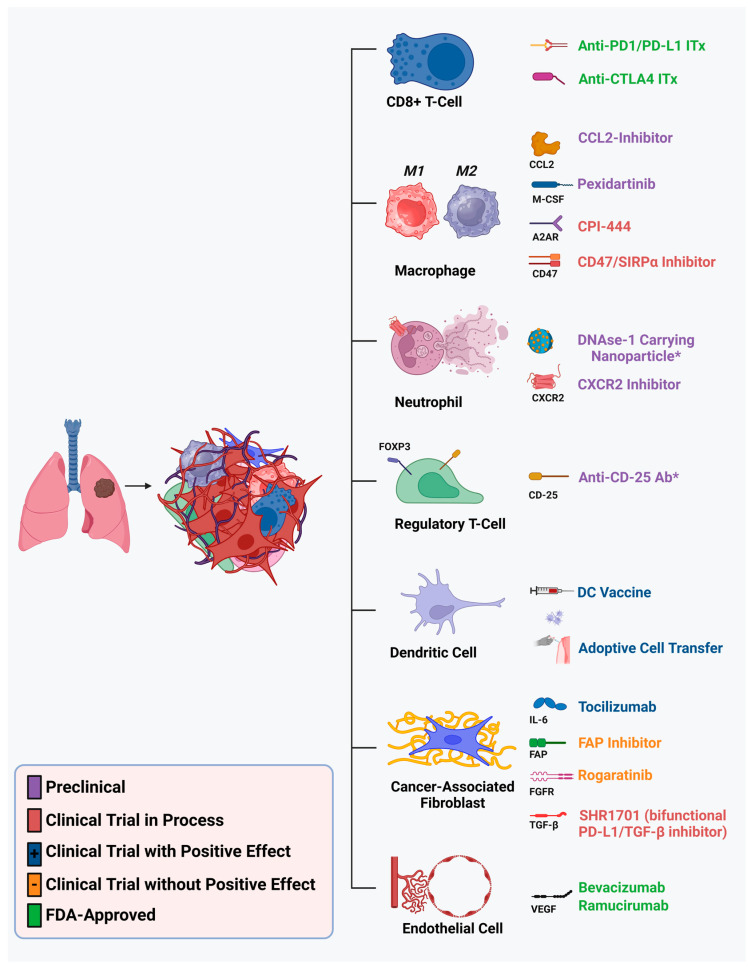
Summary of major, recent, TME component-specific therapeutic targets stratified by preclinical or clinical phases of investigation. * Investigations in other solid organ tumor models.

**Figure 2 cancers-17-01732-f002:**
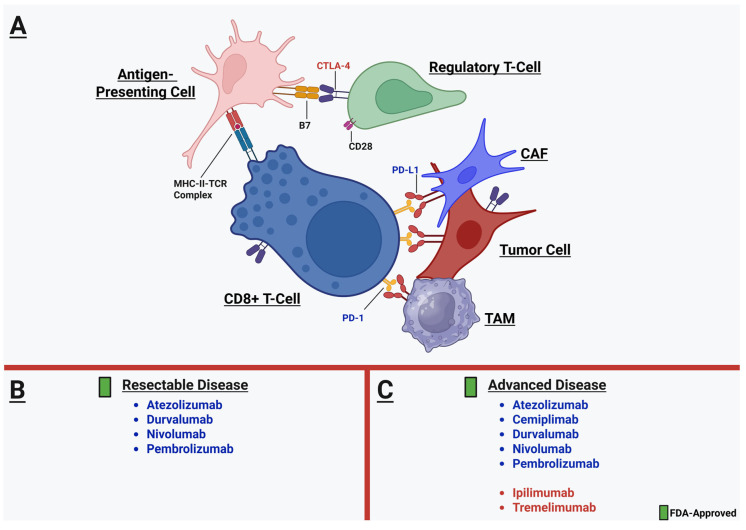
(**A**) Therapeutically targetable immune checkpoint mechanisms in the NSCLC TME, including CTLA-4-B7 (red) and PD-1/PD-L1 (blue) pathways. Notably, CTLA-4 and PD-L1 are expressed by multiple cell types in the TME, of which some key examples are depicted. Summarized below are the currently FDA-approved ICI therapies in the perioperative setting for resectable disease (**B**) and for advanced disease (**C**).

**Table 1 cancers-17-01732-t001:** Major knowledge gaps identified with respect to emerging TME-specific therapies for NSCLC.

Key Knowledge Gaps
• TME characteristics modulating ICI efficacy and biomarkers for these factors. • Identifying the mechanisms and biomarkers of immunotherapy resistance related to the TME in the majority of treated patients. • Intra- and inter-tumoral heterogeneity of the TME, its unique molecular composition at the primary tumor and metastatic sites, and its impact on ICI therapy responsiveness. One example is understanding the role of different classes of macrophages in the TME.
• Results from utilizing new single-cell analysis approaches (e.g., spatial transcriptomics, Sc-RNAseq) for identification of unique TME compositions to guide targeted therapies. • Exploration of the dense network of cytokine and signaling pathways between tumor and TME cells and identifying therapies that focus on specific pathologic pathways while avoiding interference with physiologic signaling.
• Impact of tumor oncogenotype on immunotherapy responsiveness and why certain mutational profiles (i.e., EGFR-mutant NSCLC) are more resistant. Molecular mechanism(s) underlying ICI efficacy and resistance in different NSCLC oncogenotypes.
• Role of ICI differences between administration in neoadjuvant vs. adjuvant settings. • Addressing toxicity from neoadjuvant immunotherapy for resectable disease. • Impact of neoadjuvant ICI on surgical decision-making and patient safety related to surgical procedures. • Paucity of Treg-specific therapies in the preclinical and clinical spheres.
• Role of combination therapies of ICI therapy + TME-specific targeted therapy.
• Role of the TME on ICI and radiation therapy-induced abscopal effects. • Potential impact of TME targeted therapy on rates of autoimmune pneumonitis and interstitial lung disease. • Exploring TME-specific targets to ameliorate ICI-associated toxicity.

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
