# Peer review of "The Current State of Tumor Microenvironment-Specific Therapies for Non-Small Cell Lung Cancer"

_cancers, 2025, doi:10.3390/cancers17111732_

Round 1
Reviewer 1 Report
Comments and Suggestions for Authors
In this manuscript authors address the important subject of the microenvironment in non-small cell lung cancer. This study is very well presented, with excellent selection of content and layout for the figure .
It is correctly noted in the manuscript that use of ICI has the potential for significant side effects, and possibly fibrosis. It would therefore help to identify additional methods to interfere with the ICI-driven side effects and potentiate their impact. The group of Terashima in Tokyo identified NUP85 as a prospective target for interference with macrophage activity and fibrosis, in a few articles dating 2020 and later.
In 2019, it was suggested that bromodomain inhibitors interfering with NFkB-driven gene expression have a relevance in NSCLC, due to the effects of cytokines and chemokines in shaping the NSCLC microenvironment (it was suggested to occur in part by oxidative-stress driven effects on the dual function protein OGG1). It might help to add a comment about potential ways to interfere with cytokine/chemokine expression as a means to interfere with NSCLC shaping the tumor microenvironment. Signal pathways that are activated in innate immunity interfere substantially with T-cell physiology and affect adaptive immunity, especially through potent effects on T-helper cell phenotypes.
It might be beneficial for the impact of the manuscript to add one more figure to illustrate the effects of NSCLC in shaping the microenvironment drastically, as certain cytokines have potent effects on tissue function.
Reviewer 2 Report
Comments and Suggestions for Authors
The work presented is comprehensive and registers the use of current literature.
My personal ideas about this review are:
### I think it will be very beneficial to define THERAPY and its GOALs. Can we expect all the therapies mentioned here to have the same effect? Prevent metastasis, decrease tumor volume, eliminate the tumor, etc.?
### Figure 1 seems comprehensive. Is there an explanation as to what is commonly found in the TME in NSCLC? Are these therapies all employed or under studies to be employed in NSCLC tumors? Are these therapies to enahnce the immune system but not against the TUMOR itself?
### Is there a study in the literature that examined the effects of therapy on TME cells and structures? for example, if you administer a therapeutic that affects the tumor directly, what are the consequences to the TME?
### Can you comment on the TME cells and structures? Can you provide evidence if these cells are in their physiological state or are experiencing changes compared to the healthy counterpart cells?
You mentioned intra-tumoral cells. Where do these cells come from? Is there migration of cells into the tumor? We often think about cancer cells migrating to break the extracellular matrix or even intravasating. Is this phenomenon of the endothelial cells and fibroblasts, and perhaps other cells, a two-way street? Are they engulfed by the tumor, or is there a point at which the tumor welcomes invaders (becomes more permeable)?
There are some formatting mistakes, line 68, 611 are examples. Review the entire document.
Reviewer 3 Report
Comments and Suggestions for Authors
The review “The Current State of Tumor Microenvironment-Specific Therapies for Non-Small Cell Lung Cancer” is a rather short but a well-documented review in the field. The main problem here is a lot of minor typos and flaws. I've summarized the most striking ones below. However, I'm afraid, this is not a guarantee that some remained unnoticed. I recommend for the authors to carefully proofread the manuscript.
Minor comments:
- line 68 “can is ob-“, -à can be obtained;
- lines 97-99: “Wu et al. analyzed advanced NSCLC tumors from 42 patients and 90,406 cells and identified 11 major cell types including cancer cells, non-cancer epithelial cells, immune cells, and stromal cells.” Please, insert reference to the work of Wu et al. in the end of this sentence.
- line 108: “CP2E” is this the same as CP2E? If yes, please use the same abbreviations throughout the article;
- line 165: “(Sandler et al., 2006) (Rosell et al., 2012; Rosell et al., 2017)” different style of referencing, please use reference numbers in squared brackets, as elsewhere;
- line 191: “[33, 34] [35]”, better: [33-35];
- line 232: “[51, 52] [53] [54] [55]”, better: [51-55];
- line 233: “[51] [25] [25, 56]”, better: [25,51,56];
- line 243: “Kargl et al., 2019; Yao et al., 2013).”, please, convert to numbers;
- line 374,389: “CD8”, CD8+?
- line 396: Is this section 3.2.6? If yes, please, insert this number;
- Line 607: Not the full list, missing: Tregs, NKG2D, CTLA-4+ etc. MDPI is not necessary here. Please, check other abbreviations carefully and add them to this Table;
- Line 611: extra number 1;
- References 87-92, 142-145. Please, add web-site or doi.
Reviewer 4 Report
Comments and Suggestions for Authors
Editors/confidential:
In general, I am disappointed by what could have been a very good and important review, particularly with the high reputation of the senior author, but it is an ambitious topic, and many detailed mechanistic explanations or even speculations have been omitted, which dampens enthusiasm and limits the value of the review. Please refer to the comments to the authors for more detailed suggestions. I have purposely saved time by refraining to comment on grammar and wording changes that I recommend, since you have already communicated a decision of “major revision” to the authors.
- In most of this paper, all of the histologies and subsets of NSCLC are lumped together. There are important differences in non-smoker-associated subsets and in those with various oncogenic drivers (EGFR and the less common ALK, ROS1 and others) that affect the TME.
- The authors’ selection of clinical data to include is suboptimal. One particular example is to cite in lines 180-183 a very small phase II study of bev and atezo with the ORR and the 1 yr OS implicated as being results of the therapy, but in the absence of randomization or a larger study and more details re: the patient characteristics, this is a meaningless statement.
- Actual errors are scattered throughout. In paragraph lines 100-114, there are terminology errors from the reference cited (Bischoff) to the wording used here (CPE2 and N3MC); also, the statement in line 373 that CTLA4 binds to CD28 is incorrect, and in the next line, the mechanism of co-stim inhibition by CTLA4 is misleading (attributed to CD4+ T cells not otherwise specified, while the more predominant expressors of costimulatory B7.1 are APCs and tumor cells).
- The remainder of the above paragraph and the following paragraph are confusing and would be better presented as a figure.
Authors:
- Many grammatical and word usage suggestions could be made to enhance the readability of this manuscript but are left to the authors to edit carefully and perhaps for the editorial office.
- In a discussion of TME in a group of malignancies that have different characteristics depending on driver mutations in some cases, it would be helpful to be more detailed about the subsets with different driver mutations or at least those in the majority of non-smoker adenos with EGFR mutations.
- Check line 139 which says P73 but probably means P53 and appears to cite the wrong reference.
- TGF beta is transforming not tumor in line 218.
- Whenever terms like “improved” such as in line 296 and other locations are used, they need to include a comparator. Improved over what?
- In a review of TME, it seems that you might mention CPS in addition to TPS when discussing PD-L1 expression.
- The mechanisms of EGFR mutation should be detailed as much as possible, as it’s probably the only opportunity to expound on the many studies that have been done to understand the overall favorable prognosis of these tumors, their lack of association with smoking, their very low TMB, and their very high resistance to immunotherapy following targeted Rx (but nevertheless an approval in that setting). This is where you can really address the tumor: TME crosstalk and even contrast it with different NSCLC histologies as well as subsets of adeno.
- Paragraph lines 456 to the end needs work: first, need to distinguish anti-angiogenic Rxs that are Abs like bev and ramucirumab from those TKIs (many of which are dirty kinase inhibitors) that are implicated in anti-angiogenesis and may not have the same activity in combination Rx for NSCLC. In addition to correcting the word “ralted” [does this mean “altered”?), please provide a concluding sentence to this paragraph about where these regimens are most encouraging and where they are going in the near future.
- Paragraph lines 540-547 should be removed, as it is not really relevant.
- IDO does not stand for “interferon response”—it is indoleamine dioxygenase, line 552.
Round 2
Reviewer 3 Report
Comments and Suggestions for Authors
The manuscript has been improved. However, some issues remain:
- Legend to Figure 2: Does the sentence "Summarized below are the currently FDA-approved ICI therapies in the perioperative setting for resectable disease and for advanced disease." belong to the legend? Also, I do not understand, why ipilimumab, tremelimumab are in red color, while other drugs in blue. Can it be explained in the legend?
- Line 293: [70] [38] [38, 75]. Two times ref. 38. One is enough.
- References 176-179. Make them similar to ref. 109-115, i.e. provide web-site.
Reviewer 4 Report
Comments and Suggestions for Authors
thanks for many improvements
the only residual thing to fix is line 242 "disease control rate" is not a defined or validated endpoint. need to either detail how it was defined in the quoted study or stick with just the ORR/OS/PFS which are all fully-defined endpoints.
